# Automatic Malignant and Benign Skin Cancer Classification Using a Hybrid Deep Learning Approach

**DOI:** 10.3390/diagnostics12102472

**Published:** 2022-10-12

**Authors:** Atheer Bassel, Amjed Basil Abdulkareem, Zaid Abdi Alkareem Alyasseri, Nor Samsiah Sani, Husam Jasim Mohammed

**Affiliations:** 1Computer Center, University of Anbar, Al-Anbar 31001, Iraq; 2Center for Artifical Intelligence Technology, Faculty of Information Science and Technology, Universiti Kebangsaan Malaysia, Bangi 43600, Selangor Darul Ehsan, Malaysia; 3ECE Dept., Faculty of Engineering, University of Kufa, Najaf 54001, Iraq; 4College of Engineering, University of Warith Al-Anbiyaa, Karbala 63514, Iraq; 5Information Technology Research and Development Centre, University of Kufa, Najaf 54001, Iraq; 6Department of Business Administration, College of Administration and Financial Sciences, Imam Ja’afar Al-Sadiq University, Baghdad 10001, Iraq

**Keywords:** skin cancer, deep learning, CNN, machine learning, prediction

## Abstract

Skin cancer is one of the major types of cancer with an increasing incidence in recent decades. The source of skin cancer arises in various dermatologic disorders. Skin cancer is classified into various types based on texture, color, morphological features, and structure. The conventional approach for skin cancer identification needs time and money for the predicted results. Currently, medical science is utilizing various tools based on digital technology for the classification of skin cancer. The machine learning-based classification approach is the robust and dominant approach for automatic methods of classifying skin cancer. The various existing and proposed methods of deep neural network, support vector machine (SVM), neural network (NN), random forest (RF), and K-nearest neighbor are used for malignant and benign skin cancer identification. In this study, a method was proposed based on the stacking of classifiers with three folds towards the classification of melanoma and benign skin cancers. The system was trained with 1000 skin images with the categories of melanoma and benign. The training and testing were performed using 70 and 30 percent of the overall data set, respectively. The primary feature extraction was conducted using the Resnet50, Xception, and VGG16 methods. The accuracy, F1 scores, AUC, and sensitivity metrics were used for the overall performance evaluation. In the proposed Stacked CV method, the system was trained in three levels by deep learning, SVM, RF, NN, KNN, and logistic regression methods. The proposed method for Xception techniques of feature extraction achieved 90.9% accuracy and was stronger compared to ResNet50 and VGG 16 methods. The improvement and optimization of the proposed method with a large training dataset could provide a reliable and robust skin cancer classification system.

## 1. Introduction

The goal of detecting and curing cancer in humans is a difficult one for medical science. In the United States, skin cancer is the most frequent type of cancer. Melanoma is one of the fastest-growing and most dangerous cancers. In its advanced stages, treating this cancer is very difficult. The goal of early identification and treatment of this form of cancer is to reduce the number of cancer patients in the United States. Malignant melanoma is the deadliest kind of skin cancer, accounting for 5000 fatalities per year in the United States [1,2]. Early detection of the type of cancer is particularly critical, because patients’ health problems are becoming more severe as time passes. Melanoma begins with the formation of cytes in the pigment melanin, which gives the skin its color. It has the ability to travel to the lower layers of our skin, enter the circulation, and then spread to other regions of our bodies.

Computer-assisted technologies and methods are required for early skin cancer diagnosis and detection. The accuracy of clinical diagnosis for cancer detection is improved by computer-aided techniques and equipment. The most significant non-invasive method for detecting malignant, benign, and other pigmented skin cancers is dermoscopy [3]. The eye-based examination and recording of color changes in the skin are the traditional methods of melanoma detection and main feature identification. This classic technique for skin cancer detection relies on the surface structure and color of the skin. Dermoscopy allows for improved classification of cancer types based on their appearance and morphological characteristics [4]. Dermatologists rely on their experience while inspecting dermoscopy photos. Computerized analysis of dermoscopy pictures has become an important study topic to decrease diagnostic mistakes due to the complexity and subjectivity of human interpretation [5]. The accuracy of skin cancer diagnosis can be improved by using dermoscopy pictures to identify cancer. Figure 1 shows a graphical illustration of the distinctions between melanoma and benign skin cancer.

Extensive efforts have been made to study the categorization of malignant and benign cancers based on computer image analysis algorithms to address this challenge. By merging disciplines, including image processing, computer vision, and machine learning, these systems apply a range of techniques to segmentation, detection, and classification of melanoma [7]. Inadequate data and a lack of variety in skin cancer courses are the primary issues with these studies. The conventional technique of categorization, which relies on human observation, can be flawed. 

The intelligent fault imaging-based skin cancer detection technique is sometimes a valuable tool for assisting a physician in skin cancer classification. Dermoscopy is a noninvasive imaging technology used to enhance the process of cancer diagnosis. Dermoscopy allows for the acquisition of a magnified and lighted picture of the affected region of the skin in order to improve the clarity of the skin mark.

Researchers have proposed and tested machine learning algorithms for benign and malignant skin cancer using KNN, SVM, clustering, regression, and deep learning approaches [8]. The texture, visual, and color set of features are concentrated for the cancer identification and classification. 

Deep learning is the most popular and robust technique for classification based on execution time, complexity, and performance of the system [9]. The conventional classification techniques were restricted to the transformation of raw input into a solution of predicted and classifier outcome results [10,11,12]. The deep learning method improves the model with the development of an automated classification and prediction system. Researchers are resorting to the creation of hybrid approaches combining deep learning techniques for performance enhancement and high accuracy-based findings [13]. Stacking is a machine learning strategy that combines Meta classifiers with other machine learning approaches. Individual classifier techniques are trained using the original dataset and derived features as a basis. Fitted output as meta characteristics are derived from the functions of separate classifiers to form the basis of the meta classifier. The number of folds for training and the level-based model increase the final predicted value and classification outcome [14]. 

The core advantage of deep learning is to train features from data without the help of human experts. The deep learning technique has greatly improved the understanding and implementation of computer vision problems [15]. Based on data from prior skin cancer identification research, there is a lot of potential for developing a deep neural network-based malignant and benign skin cancer detection system. There is no evidence of any research on the use of a diverse dermoscopy skin cancer dataset for multi-class categorization of dermoscopy skin cancer pictures.

This paper aims to design an automated classification of malignant and benign skin cancer based on the hybrid method of deep learning. The proposed method is based on stacking the classification model into one multi-class model in order to improv the prediction analysis. The proposed method was implemented for three folds of training. 

The following is how the rest of the paper is organized: The rest of this section discusses similar work. Section 2 explains the methods and materials used. The experimental findings are presented in Section 3. Section 4 includes a discussion and conclusion, as well as references.

### Related Work

Many studies on the detection and diagnosis of malignant and benign skin cancer have been conducted in the last decade. The numerous datasets are provided for the research community. Researchers have applied strategies based on splitting, merging, clustering, and classification to the identification and treatment of skin cancer. Each approach has its own set of limitations and advancements from the medical community to assist medical experts in making decisions.

Rajasekhar et al. (2020) suggested an automated melanoma detection and classification approach based on border and wavelet-based texturing algorithms. For wavelet-decomposition and boundary-series models, the suggested approach used texture, border, and geometry information. SVM, random forest, logistic model tree, and hidden naive Bayes algorithms were used to classify the data [16].

A malignant skin cancer recognition system based on a support vector machine was proposed by Murugan et al. (2019). The asymmetry, border irregularity, color variation, diameter, and texture features were used for the classification of the system. The texture of the skin is the dominant feature used for decision making. The convolution neural network using the VGG net is used for the problem solving of skin cancer identification. The system is trained using the transfer learning approach [17]. 

Seeja et al. (2019) presented the heuristic hybrid rough set particle swarm optimization (HRSPSO) method for segmenting and classifying a digital picture into multiple segments that are more relevant and easier to study [18]. 

Goyal et al. (2019) offered three classification algorithms and proposed a multi-scale integration strategy for segmentation [19]. Multiclass classification, binary classification, and an ensemble model are all examples of classification methods. Taghanaki et al. (2020) employed the discrete wavelet transform to extract features and analyze texture. These collected characteristics were then used to train and assess the lesions as malignant and benign using stack auto encoders (SAEs) [20].

The early diagnosis of cancer categorization based on interpretation, according to Hasan, Md Kamrul et al. (2020), is time-consuming and subjective. Adaptive threshold, gradient vector flow, adaptive snake, level set technique, expectation-maximization level set, fuzzy based split, and merging algorithm were among the six segmentation methods employed in the suggested system. The system’s performance is measured using four different metrics: HM, TDR, FDR, and HD. The AS and EM-LS algorithms are resilient and helpful for skin cancer segmentation, according to the experimental results [21].

The various researchers have proposed different algorithms and testing features for skin cancer identification. The detailed summary of skin cancer identification is explained in Figure 2.

The steps include preprocessing, image segmentation, feature extraction, and classification. During preprocessing, the quality of the raw image is improved with the help of the removal of unwanted image information. The preprocessing steps play an important role in the performance improvement of the system. The preprocessing stages are applied using the filtering approach. For removing Gaussian noise, speckle noise, Poisson noise, and salt and pepper noise, a variety of filters can be used, including the median filter, mean filter, adaptive median filter, Gaussian filter, and adaptive wiener filter [23].

In image segmentation, the input image is divided into the various regions. The prominent and useful region of the image is highlighted and used for the further processing of feature extraction. The various methods such as Otsu, active contour, watershed, and k-mean image segmentation algorithm were used for the segmentation approach. For feature extraction, the ABCD rule is used. From this rule, the asymmetry, border, color, and diameter are the most prominent features and are extracted and passed to the classification process [24].

In addition, Nafea et al. (2021) [25], proposed a semantic method based on latent semantic analysis (LSA) to improve the adverse drug reactions reported by the patient. Three types of classification were used to propose LSA: support vector machine (SVM), naïve Bayes (NB), and linear regression (LR). Additionally, Al-Ani et al. (2021) [26], proposed a hybrid technique based on long short term memory (LSTM) and auto encoder to improve the classification of sarcasm detection through tweet reviews. Jamal et al., 2012 [27] developed a classification of Malay pantun by using support vector machines (SVM). The purpose of applying this method was to find suitable usage of poetry retrieval, according to certain contextual situations. 

Based on the evidence of previous research on the classification approach, it is observed that there is a need for a hybrid deep learning method using a stacking mechanism for malignant and benign skin cancer classification. The stacking allows the combination of the multiple classification models for training and testing. 

## 2. Materials and Methods

In this section, the dataset used for the research and proposed approach of the classification is described. 

### 2.1. Dataset for Research

For any research, data collection is one of the most important steps. The data collected using the application-based standard is the biggest challenge in machine learning. For this research, the dataset is taken from the ISIC archive [28]. The dataset consists of 1800 images of benign type and 1497 pictures of malignant cancer. Further, we have divided our dataset into two parts where 70% of the data were randomly selected for training the system and rest was used to test it. Table 1 show the details of the dataset which are used in this work.

The graphical representation of the sample images from the dataset in the two-class category for benign is represented in Figure 3. 

### 2.2. Methods for Implementation 

Over the past few years, the advancement of a convolution neural network has been developed by researchers to solve the computer vision problem more precisely within minimum time. The features were extracted using the pre-trained (Xception) model for obtaining the features of each image in the dataset [29]. The deep convolution neural networks were pre-trained using Tensor Flow. Tensor Flow is a deep learning framework developed by Google [30]. The structure of the full convolution neural network is described in Figure 4.

The implementation of deep learning mechanisms has various modules and forms. This research concentrated on auto encodes. They belong to the unsupervised learning class of neural networks. The graphical representation of the auto encoder is shown in Figure 5.

The dataset was tested using the regression, SVM KNN, RF, and deep learning techniques. Our stacking approach was compared to the performance of testing results in the literature. The experiment was conducted on a computer system with a core Intel4 processor with 12 GB RAM. A brief conceptual block diagram is illustrated in Figure 6.

#### 2.2.1. Proposed Method

The proposed method for malignant and benign skin cancer detection is designed based on the stacked CV algorithm. The meta-classification mode of machine learning is implemented by the stacked CV classifier using cross-validation. For the level-based input and to prevent over fitting the classification system, meta class classification was used. Stacking is a machine learning technique to combine multiple classification models via a meta classifier. The first-level classifiers in stacked CV are fitted to the same training set that is used to generate the inputs for the second-level classifier, which may result in overfitting. The stacked CV classifier focused on cross-validation functions in which the dataset is divided into K folds and K rounds. The first level classifier was fitted using K1 folds. The first-level classifier was employed for the remaining subset fold that was not used in the fitting-based iteration for each round [33]. The resulting prediction is stacked and fed into the second level classification as an input. The first level classifier was fitted to the total dataset in the training mode of the stacked CV classifier. The functional diagram of the stacked CV algorithm is shown in Figure 6.

**Figure 6 diagnostics-12-02472-f006:**
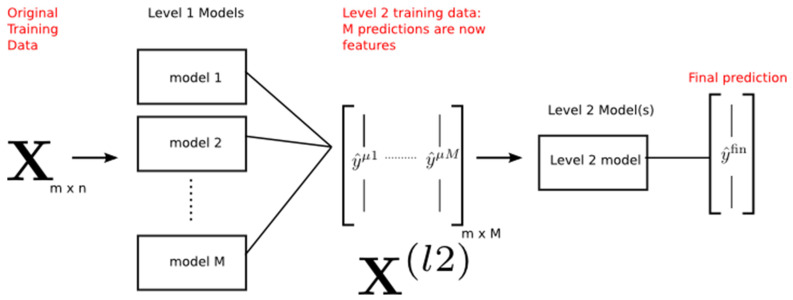
Functional diagram of Stacked CV Algorithm [34].

From Figure 6, the original training of data (X) has m observation and n features. The dimension of the original training data is m*n. M is the different model of classification that was trained on data X. Each classification model provides the prediction for the outcome results as (y). The prediction of the model is cast to the second level of training data (X12). The final results are achieved based on the final prediction model. For the design and creation of second-level data (X12), various methods are available. Stacking uses a similar idea to K-folds cross-validation to create out-of-sample predictions. The basic architecture of the stacked CV algorithm is shown in Figure 7.

Based on the functionality and basic architecture of stacked CV, the proposed block diagram for the research is shown in Figure 8.

The proposed stacking-based classification method has three folds. The original training data are passed to the level 1 model for classification such as deep learning. The outcome of the classifier of deep learning becomes prediction 1, which becomes a feature for the level 2 training data.

The level 2 training data are trained using support vector machine (SVM), neural network (NN), random forest (RF), and K- nearest neighbor (KNN) classifiers. The outcome of each classifier of the level 2 model is a prediction and it acts as a feature for the level 3 training data.

The level 3 training data are passed to the level 3 model towards the classification. The output of the level 3 model is the final prediction and used as the outcome results. The final prediction outcome detects the class of skin cancer as a result.

#### 2.2.2. Feature Extraction

The feature extraction is the most prominent and important step of the research. The purpose of feature extraction is to reduce the number of features from images of the dataset. The main aim is to create new features from the existing ones, as per their importance in the research. The new reduced set of features contains the most important information contained in the original set of features. The following factors are highlighted in the feature selection and creation process.

**Resnet50**: For this form, the convolution neural network is run with 50 layers deep. The kernel size is 7 * 7 and there are 64 different kernels with a stride of size 2 for each layer.

**VGG16**: For this form of network the VGG network uses only 3 * 3 convolution layers stacked on top of each other where Max Pooling is used to reduce volume size. It consists of 16 weight layers in which there are 13 convolution layers and 3 fully connected layers.

**Xception**: For this research, the Xception design is performed using a linear stack of depth-wise dissociable convolution layers with residual connections. It has 36 convolution layers forming the feature extraction base of the network.

This proposed design is straightforward to outline and modify. The modification is easier in the environment of Tensor Flow. The Tensor Flow takes only thirty to forty lines of code employing a high-level library. Its implementation is available as an open-source by MIT license.

### 2.3. Classification Approach

The classification of this research is performed using the stacked CV model. The classifications are extended with the support vector machine, KNN, RF, decision tree, and regression. The comparative analysis is conducted with the proposed approach and other traditional techniques used for the classification.

### 2.4. Performance Evaluation Metrics

For the performance evaluation of this experiment, the accuracy, sensitivity, F1 score, and AUC score are used. True positive (*TP*) is the correct classification of the positive class and true negative (*TN*) is the correct classification of the negative class. False positive (*FP*) is the incorrect prediction of the positives class and false negative (*FN*) is the incorrect prediction of the negatives class. The receiver operating characteristic curve (ROC-curve) corresponds to the performance of the proposed model at all classification thresholds. It is the graph of the true positive rate vs. false positive rate [36,37].
$$TPR=\frac{TP}{TP+FN}\;FPR=\frac{FP}{FP+TN}$$

AUC provides the area under the ROC-curve integrated from (0, 0) to (1, 1). It gives the aggregate measure of all possible classification thresholds. AUC has a range of 0 to 1. A 100% correct classified version will have the AUC value 1.0 and it will be 0.0 if there is a 100% wrong classification. The F1 score is calculated based on precision and recall. The mathematical representation of precision and recall are explained below [38,39].

Precision checks how precise the model works by checking the correct true positives from the predicted ones.
$$\mathrm{Pr}ecision=\frac{TP}{TP+FP}$$

Recall calculates how many actual true positives the model has captured, labeling them as positives.
Recall=TPTP+FNF1=2×Precision×RecallPrecision+Recall

The accuracy is the most important performance measure. Accuracy determines how many true positives *TP*, true negatives *TN*, false positives *FP*, and false negatives *FN* were correctly classified [39,40,41].
$$Accuracy=\frac{TP+TN}{TP+TN+FP+FN}$$

The sensitivity is the performance measure, and it is calculated as the number of positive items correctly identified.
$$Sensitivity=\frac{TP}{TN+FN}$$

## 3. Experimental Analysis

The experiment is tested with three modes of the feature extraction: Resnet50, Xception, and VGG 16. From the extracted feature the system is passed through the classification mode of SVM, KNN, regression, AdaBoost, RF, decision tree, and GaussianNB. The system is tested with our proposed stacking approach which is a hybrid combination of the proposed model. This proposed approach aims to improve the classification performance of the system. This research splits 70% of the dataset as a training set, 15% as a validation set, and 15% as the testing set to evaluate the performance. For the evaluation of the performance of the system, the accuracy, F1 score, sensitivity, and area under ROC (AUC) metrics are used. The numerical outcome of the Resnet50 features with the performance evaluation metrics is described in Table 2. The graphical representation of the comparative performance of Resnet50 features with a given classification approach is shown in Figure 9.

The numerical results of the Xception features with the performance evaluation metrics are described in Table 3. The graphical representation of the comparative performance of Xception features with a given classification approach is shown in Figure 10.

The experimental outcome results of the VGG16 features with the performance evaluation metrics are described in Table 4. The graphical representation of the comparative performance of VGG16 features with a given classification approach is shown in Figure 11.

The comparative performance of the system with all classification approaches is calculated in Table 5. The graphical representation of the comparative performance of the system is shown in Figure 12.

The ROC curve and area under the ROC curve is the most prominent results for the performance evaluation. The graphical representation of the ROC curve of this research is shown in Figure 13.

The performance testing based on the proposed model was utilized by the researcher. In this analysis, the classification of malignant and benign cancer was performed using the stacking CV model implemented using a deep learning approach. The experiment was tested in a three-fold training mechanism. The original dataset was trained using a deep learning approach. The output of deep learning became a feature set for the level 2 model such as with SVM, RF, NN, and KNN techniques. The second level utilized the prediction of the previous classifier as output and processed the prediction. The prediction was the third level model in the stacking CV algorithm and was extracted based on the previous level output. For the proposed approach, stacking CV on the Xception feature extraction mode proved dominant and promising, with 90.9% accuracy.

## 4. Conclusions

This research proposed a hybrid deep learning approach for the classification of a system. For this research, the dataset was taken from the ISIC archive. The dataset consisted of 1800 images of benign type and 1497 pictures of malignant type cancer. The experiment was tested with three modes of the feature extraction: Resnet50, Xception, and VGG 16. From the extracted feature the system was passed through the classification method of deep learning, SVM, KNN, NN, regression, and random forest.

This research proposed a Stacking CV approach for the classification of malignant and benign skin cancer based on meta classification. The system was tested with our proposed stacking approach, which is a hybrid combination of deep learning with the stacking mechanism. This proposed approach aims to improve the classification performance of the system. This research splits 70% of the dataset as a training set, 15% as the validation set, and 15% as a testing set to evaluate the performance. The performance of the system is evaluated based on accuracy, F1 score, sensitivity, and AUC score metrics. The proposed approach of stacking CV on the Xception feature extraction mode proved dominant and promising with90.9% accuracy. More diverse datasets with varied categories and different ages and more dermoscopy images with balanced samples per class are needed for further improvement. Additionally, using the metadata of each image can be useful to increase the accuracy of the model.

## Figures and Tables

**Figure 1 diagnostics-12-02472-f001:**
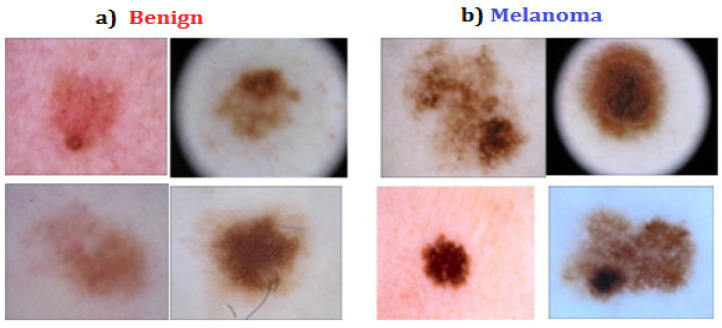
Representation for (**a**) “benign cancer” (**b**) “Melanoma” [6].

**Figure 2 diagnostics-12-02472-f002:**
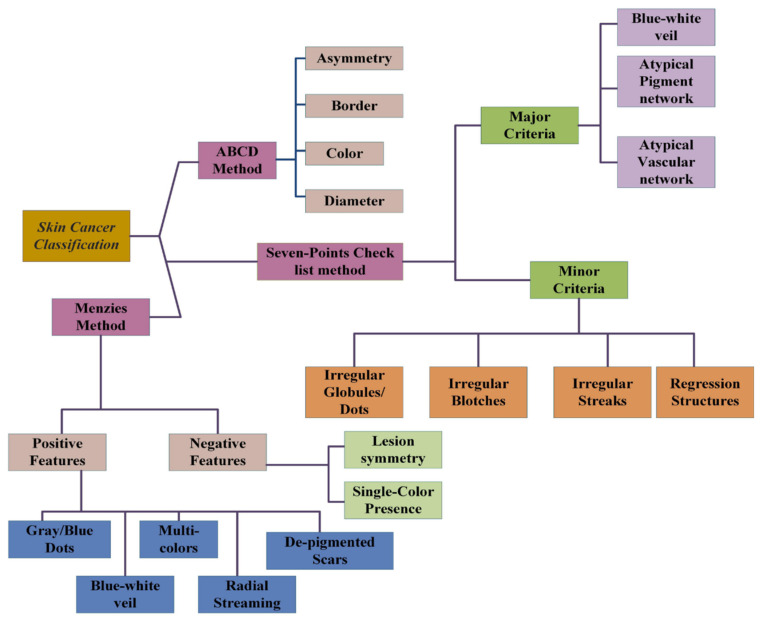
Summary used for skin cancer identification [22].

**Figure 3 diagnostics-12-02472-f003:**
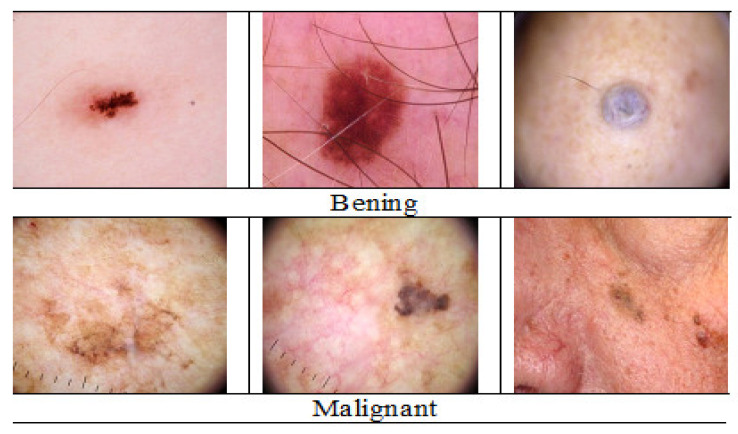
Samples for benign and malignant cancer in the dataset.

**Figure 4 diagnostics-12-02472-f004:**
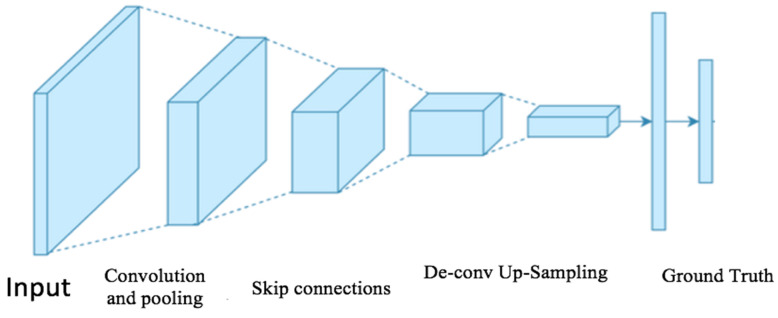
Structure of convolution neural network [31].

**Figure 5 diagnostics-12-02472-f005:**
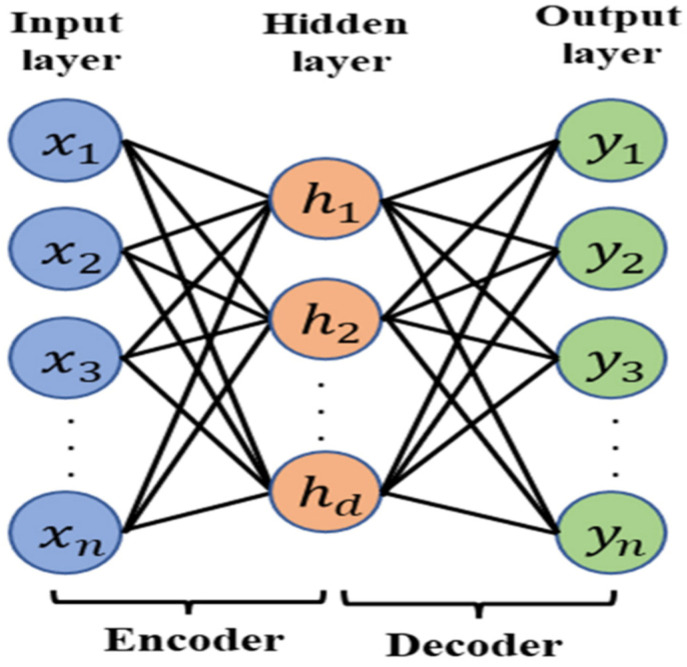
Auto encoder module for convolution neural network [32].

**Figure 7 diagnostics-12-02472-f007:**
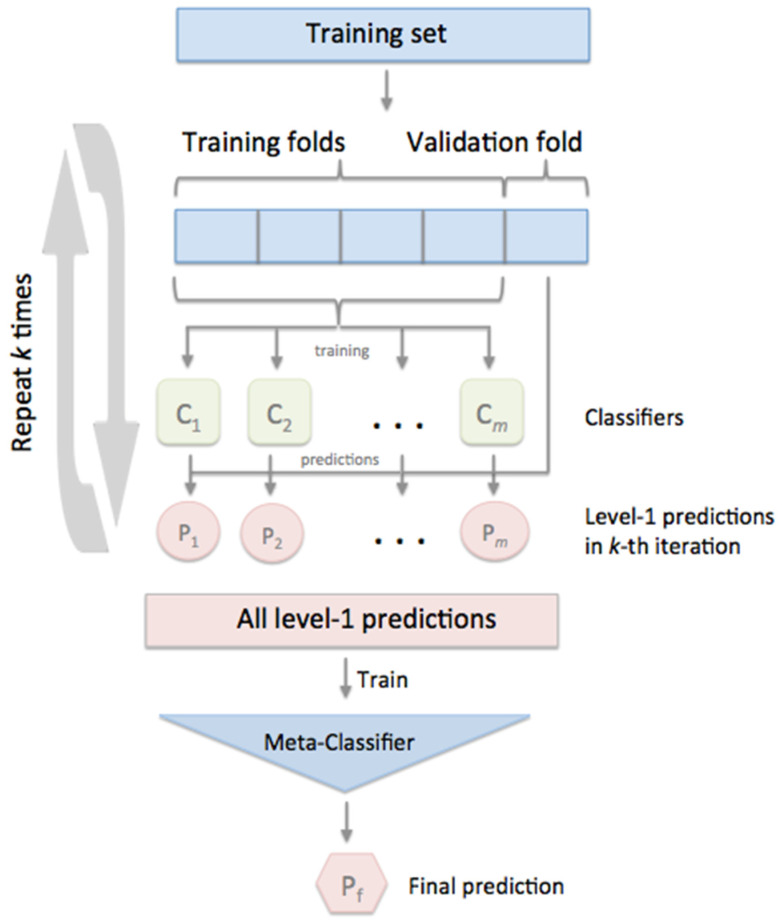
The basic architecture of stacked CV algorithm [35].

**Figure 8 diagnostics-12-02472-f008:**
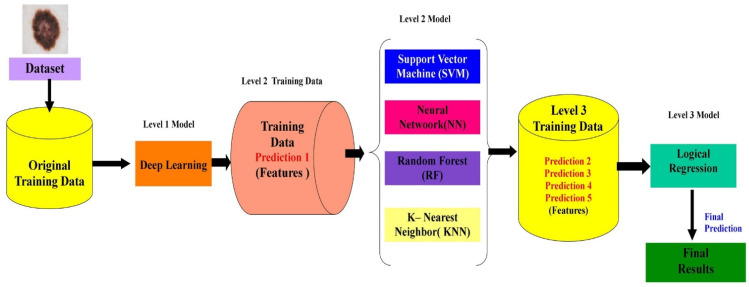
Proposed stacked CV based block diagram of the research.

**Figure 9 diagnostics-12-02472-f009:**
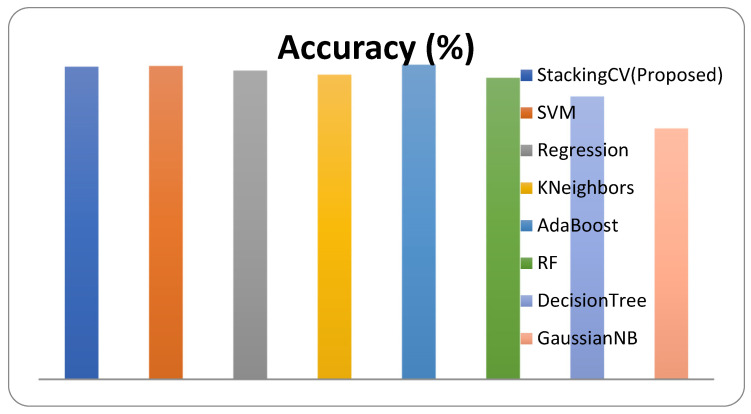
Performance of the system for Resnet50 features.

**Figure 10 diagnostics-12-02472-f010:**
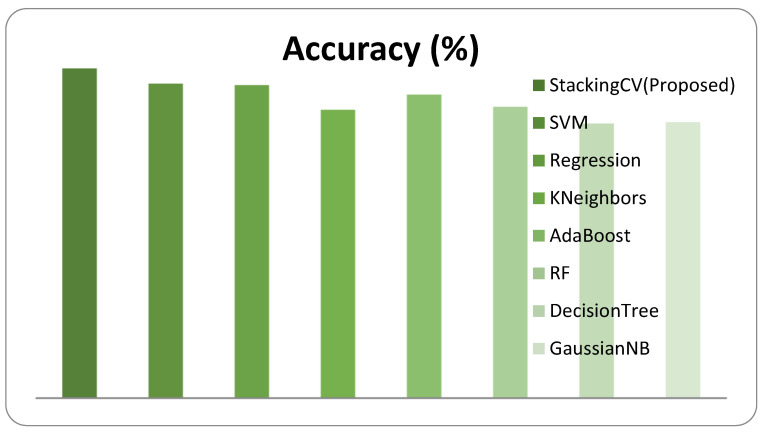
Performance of the system for Xception features.

**Figure 11 diagnostics-12-02472-f011:**
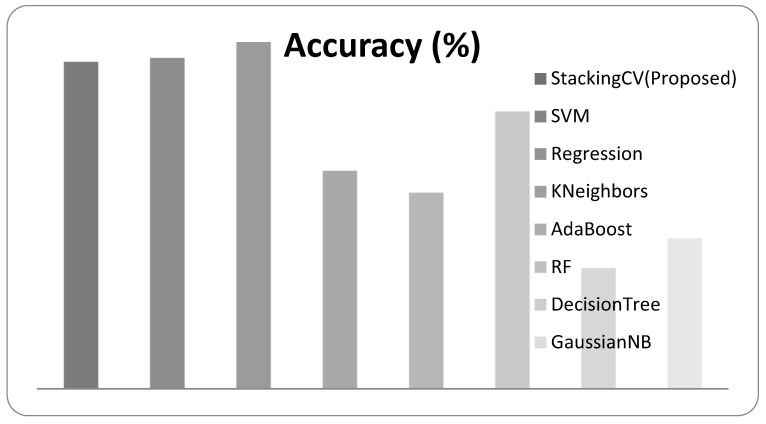
Performance of the system for VGG16 features.

**Figure 12 diagnostics-12-02472-f012:**
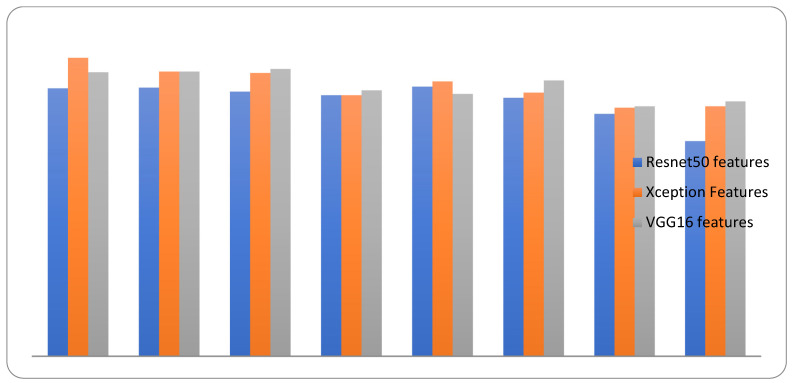
Comparative performance of the proposed and available classification approaches.

**Figure 13 diagnostics-12-02472-f013:**
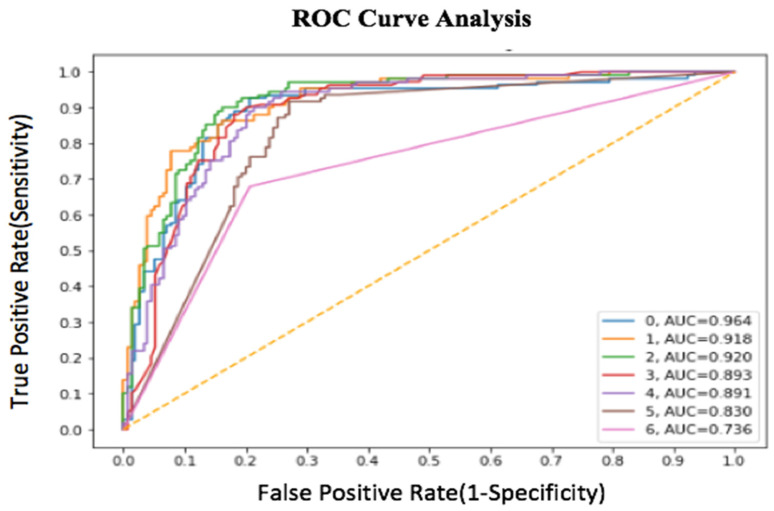
ROC curve analysis of the proposed research.

**Table 1 diagnostics-12-02472-t001:** ISIC dataset used for the research.

Skin Cancer Class	Number of Images
Benign	1440 + 360
Malignant	1197 + 300

**Table 2 diagnostics-12-02472-t002:** Performance of the system for Resnet50 feature extraction.

Classifier	Accuracy (%)	F1-Score	Sensitivity	AUC
StackingCV (Proposed)	81.6	0.788	0.821	0.818
SVM	81.8	0.787	0.816	0.817
Regression	80.6	0.762	0.752	0.798
KNN	79.5	0.754	0.761	0.790
AdaBoost	82.1	0.792	0.825	0.822
RF	78.7	0.735	0.715	0.777
DecisionTree	73.8	0.666	0.633	0.722
GaussianNB	65.5	0.363	0.238	0.593

**Table 3 diagnostics-12-02472-t003:** Performance evaluation of the system for the **Xception** feature extraction method.

Classifier	Accuracy (%)	F1-Score	Sensitivity	AUC
StackingCV (Proposed)	90.9	**0.896**	**0.886**	**0.917**
SVM	86.7	0.838	0.834	0.862
Regression	86.3	0.837	0.853	0.862
KNN	79.5	0.732	0.678	0.778
AdaBoost	83.7	0.801	0.798	0.831
RF	80.3	0.739	0.678	0.784
DecisionTree	75.7	0.676	0.614	0.736
GaussianNB	76.1	0.731	0.788	0.765

**Table 4 diagnostics-12-02472-t004:** Performance evaluation of the system for **VGG16** feature extraction.

Classifier	Accuracy (%)	F1-Score	Sensitivity	AUC
StackingCV (Proposed)	86.5	0.842	0.804	0.843
SVM	86.7	0.835	0.810	0.859
Regression	87.5	0.847	0.844	0.870
KNN	81	0.761	0.733	0.799
AdaBoost	79.9	0.766	0.798	0.799
RF	84	0.805	0.798	0.834
DecisionTree	76.1	0.701	0.678	0.749
GaussianNB	77.6	0.723	0.706	0.766

**Table 5 diagnostics-12-02472-t005:** The comparative performance based on the accuracy.

Classifier	Resnet50 Features	Xception Features	VGG16 Features
StackingCV (Proposed)	81.6	**90.9**	86.5
SVM	81.8	86.7	86.7
Regression	80.6	86.3	87.5
KNeighbors	79.5	79.5	81
AdaBoost	82.1	83.7	79.9
RF	78.7	80.3	84
Decision Tree	73.8	75.7	76.1
GaussianNB	65.5	76.1	77.6

## Data Availability

The data presented in this study are available in article.

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
