# Peer review of "Automatic Malignant and Benign Skin Cancer Classification Using a Hybrid Deep Learning Approach"

_diagnostics, 2022, doi:10.3390/diagnostics12102472_

Round 1

Reviewer 1 Report

" Automatic Malignant and Benign Skin Cancer Classification Using a Hybrid Deep Learning Approach "

It is very interesting to design an automated classification of Malignant and Benign skin cancer based on the hybrid method of deep learning. However, there are a few corrections that are essential to meet the standard for publication. Please refer to the following comments.

1)    This study is very interesting, but more details about the study design are needed. How did you ensure internal validity in datasets other than StackingCV? Please explain how to manage each fold of the dataset. Is it cross-validation? Leave out method?

2)    Please examine the misclassified data in detail. StackingCV is great, but did the misclassification content differ? Consider content as well as precision numbers.

3)    Did the feature regions differ for each CNN model? Please add feature region visualization.

Author Response

Thank you for your comments, please check the response in the attached file.

Reviewer 2 Report

The authors present a paper about "Automatic Malignant and Benign Skin Cancer Classification Using a Hybrid Deep Learning Approach".

The topic is interesting and several rearchers have already provided valuable contributions about it.

There are several points that should be implemented as follows:

1) which is the target of the proposed model? (priamry care doctors? dermatologists? patients?) please explain the potential clinical use

2) The feature choice is probably the most significant part when dealing with artificial intelligence: which were the features analyzed? how were they chosen? were they different from others models presetned in section 1.1?

3) Is 90.9% accuracy according to the authors a result which could be tranlated into clinical practice? if not how could it be implemented? 

4) it would be interesting to test the Hybrid Deep Learning Approach proposed by the authors against an expereinced dermatologist to comapre the results 

Author Response

(The authors gave the same response as above.)

Round 2

Reviewer 1 Report

Thank you for giving me this opportunity to re-review your revised manuscript. 

I am happy that all of the suggested corrections have been made.

Thank you for spending so much time for revised manuscript.

Reviewer 2 Report

I have no further comments